# Toll-Like Receptor as a Potential Biomarker in Renal Diseases

**DOI:** 10.3390/ijms21186712

**Published:** 2020-09-13

**Authors:** Sebastian Mertowski, Paulina Lipa, Izabela Morawska, Paulina Niedźwiedzka-Rystwej, Dominika Bębnowska, Rafał Hrynkiewicz, Ewelina Grywalska, Jacek Roliński, Wojciech Załuska

**Affiliations:** 1Department of Clinical Immunology and Immunotherapy, Medical University of Lublin, 20-093 Lublin, Poland; izabelamorawska19@gmail.com (I.M.); jacek.rolinski@gmail.com (J.R.); 2Department of Genetics and Microbiology, Institute of Microbiology and Biotechnology, Faculty of Biology and Biotechnology, Maria Curie-Skłodowska University, Akademicka 19 St., 20-033 Lublin, Poland; paulina.lipa56@gmail.com; 3Institute of Biology, University of Szczecin, Felczaka 3c, 71-412 Szczecin, Poland; bebnowska.d@wp.pl (D.B.); rafal.hrynkiewicz@gmail.com (R.H.); 4Department of Nephrology, Medical University of Lublin, 20-954 Lublin, Poland; wojciech.zaluska@umlub.pl

**Keywords:** acute kidney injury, biomarker, diabetic nephropathy, focal segmental glomerulosclerosis, innate immunity, membranous nephropathy, minimal change diseases, TLR

## Abstract

One of the major challenges faced by modern nephrology is the identification of biomarkers associated with histopathological patterns or defined pathogenic mechanisms that may assist in the non-invasive diagnosis of kidney disease, particularly glomerulopathy. The identification of such molecules may allow prognostic subgroups to be established based on the type of disease, thereby predicting response to treatment or disease relapse. Advances in understanding the pathogenesis of diseases, such as membranous nephropathy, minimal change disease, focal segmental glomerulosclerosis, IgA (immunoglobulin A) nephropathy, and diabetic nephropathy, along with the progressive development and standardization of plasma and urine proteomics techniques, have facilitated the identification of an increasing number of molecules that may be useful for these purposes. The growing number of studies on the role of TLR (toll-like receptor) receptors in the pathogenesis of kidney disease forces contemporary researchers to reflect on these molecules, which may soon join the group of renal biomarkers and become a helpful tool in the diagnosis of glomerulopathy. In this article, we conducted a thorough review of the literature on the role of TLRs in the pathogenesis of glomerulopathy. The role of TLR receptors as potential marker molecules for the development of neoplastic diseases is emphasized more and more often, as prognostic factors in diseases on several epidemiological backgrounds.

## 1. Introduction

The term “biological marker” in the literature refers to the objective health status of the patient, which can be observed from the outside, i.e., in a way that can be measured in an extremely accurate and, most importantly, repeatable manner. Various medical symptoms can often contrast with each other, or, on the contrary, correlations between them are observed, which may contribute to a better and faster diagnosis of the disease or the effectiveness of the treatment process. A joint project of the World Health Organization together with the United Nations under the name of the International Program on Chemical Safety has attempted to standardize the term biomarker used in the literature. Currently, a biomarker is defined as “any substance, structure or process that can be measured in the body or its products and that can influence or predict the occurrence of an outcome or disease” [1], which means that the modern definition of biomarkers covers every response of the body, whether functional, physiological, or biochemical, to the occurrence of a potential threat (physical, chemical, or biological) that may modify the body’s reactions at the cellular or molecular level. Usually, biomarker is used with two meanings: (1) A biomarker is a component (analyte) of a human biological system (i.e., plasma, urine, etc.); or (2) a biomarker is a biological property (i.e., mass concentration of X in plasma) [2]. Currently, examples of biomarkers can be both simple parameters of heart rate and blood pressure, as well as the determination of complex constituent molecules present in our blood or tissues, which are indicative of health. Types and applications of biomarkers are constantly changing as it evaluates our knowledge about them. Thus, it has recently been shown that counts of urinary T-cell, renal tubular epithelial cells, and podocalyxin-positive cells provide an excellent biomarker for the detection of renal transplant rejection in routine clinical trials [3]. Currently, the role of TLR receptors as potential marker molecules for the development of neoplastic diseases is emphasized more and more often, e.g., TLR5 as a prognostic marker for gastric cancer [4]. It was also shown that the activity of TLR receptor is correlated with the state of injury of post-surgical patients who have a disorder of the immune response related to the interaction of TLR receptors with DAMP (damage-associated molecular pattern). Moreover, the analyses led to conclusions regarding the role of TLR receptors in predicting pathological conditions, including tissue damage, in these patients [5]. One of the increasing threats in today’s world is chronic kidney disease, the cause of which may be both primary processes related to the kidneys, and secondary processes observed in the course of rheumatic, cardiological, or diabetic diseases. As indicated in the literature, one of the largest causes of the development of chronic kidney disease in highly developed countries as well as in developing countries is hypertension and type 2 diabetes, currently classified as diseases of civilization [6]. As far as type 2 diabetes and renal disorders are concerned, it was shown that polypharmacy has a great impact on the occurrence, course, and treatment of the disease [7].

Genetic, epigenetic, or environmental factors that play a more or less important role in different regions of the world may lead to diseases associated with kidney glomerular damage, which leads to their chronic hypofunction and subsequent renal failure. Research conducted by O’Shaughnessy et al. [6], aimed to analyze the data from centers dealing with chronic kidney diseases, where several types of nephropathy were considered. Analysis shows that there is a difference in the incidence of individual nephropathy depending on the region. It turned out that focal segmental glomerulosclerosis (FSGS) and diabetic nephropathy are the most numerous in the USA because it occurred in 19% of diagnosed patients; the second type that dominated was IgA nephropathy (12%), another was membranous nephropathy (12%) and lupus nephritis (10%). Furthermore, FSGS was also common in Latin America (16%), although lupus nephritis strongly dominated the region (38%), while diabetic nephropathy (DN) (4%) and IgA nephropathy (IgAN) (6%) were relatively rare. A reverse dependence than in the USA has been observed in Europe and Asia. In Europe, IgA nephropathy dominated the diagnosis, and the second most common diagnosis was FSGS (15%), while in Asia, IgA glomerulopathy included 40% of diagnoses and the second most common diagnosis was lupus nephritis (17%) [8].

All of these diseases are the subject of intensive research by many scientists around the world. The following review of the literature focuses on the problems of diseases associated with glomerular dysfunction, in which there are answers to questions about the participation in the pathogenesis of diseases on TLR, which may become a potential marker molecule.

## 2. Classification of Biomarkers

There are many different ways to classify biological molecules as biomarkers in the literature. One of the basic methods is division according to the type of marker molecule. These are DNA, mitochondrial DNA, RNA, or mRNA molecules that belong to the genomic biomarker category, but also proteins, peptides, or antibodies that are classified as proteomic biomarkers. The third group is metabolic markers (metabolomics), which includes lipids, carbohydrates, enzymes, and products of metabolism (metabolites). Our attention was drawn to a slightly different classification of these unusual molecules, i.e., classification based on genetics and molecular biology, due to the usefulness of biomarkers in diagnostic processes and due to their application (Figure 1). The first group includes three types of biomarkers. Type 0 are biomarkers that correlate over time with known clinical indications and show the natural course and history of the disease. Type I relates to biomarkers of drug activity, which can be divided into biomarkers of efficacy (taking into account the therapeutic effect of a given drug), mechanism (providing information about the drug’s mechanism of action), and toxicity (including the toxic effect of a given drug). Type II is known as surrogate biomarkers to help evaluate and predict the effect of therapy [9].

Due to the usefulness of biomarkers in diagnostics, we can distinguish prognostic biomarkers; in other words, those that suggest the probable outcome of the disease in an untreated individual, and predictive biomarkers, the purpose of which is to identify patients for whom a specific therapy is most effective. Lastly, we can distinguish pharmacodynamic biomarkers that determine the pharmacological action of a given drug [10]. Another classification is based on the use of biomarkers. We distinguish here exposure markers and doses that are used to reconstruct and predict past accidental or occupational exposures. Risk or vulnerability markers, which relate to the identification of vulnerable individuals (or future patients) at increased risk of developing a disease, and disease markers represent the initial cellular or molecular changes that occur during the development of a particular disease entity. It is the latter group that includes TLR receptors [11,12]. However, what characteristics does an ideal biomarker need to meet? Is there an “ideal biomarker”? What features should a biomarker have to be close to the ideal? Well, according to the FDA, an ideal biomarker must meet the following six characteristics (Figure 2). First, it must be specific in the course of a particular disease entity and easy to differentiate between different physiological conditions of the patient. Secondly, such a biomarker must above all be easy to measure and safe. Then, the speed of its detection is also important, as it enables a quick diagnosis as well as the repeatability and accuracy of the results obtained. Attention is also paid to the cost of detecting such biomarkers, which must be relatively cheap. The advantage of such a biomarker is also the consistency between ethnic groups and genders of patients [13].

## 3. Characteristic of the TLR Receptors

Glomerulonephritis is a heterogeneous group of diseases whose common denominator is inflammation, ongoing in the glomerulus, resulting from systemic (secondary glomerulonephritis) or only glomerulonephritis (primary glomerulonephritis) [14,15,16,17,18,19]. The etiopathogenesis and the cause of the variable course of glomerulopathy is the subject of numerous studies but remains unknown. Although the pathogenesis is not unequivocally elucidated, the literature data clearly indicate the involvement of various immune mechanisms in the etiopathogenesis of glomerulopathy. Researchers indicate the role of the immune system in the development of chronic kidney disease on the basis of primary and secondary disorders of the glomerular functions, and AKI (acute kidney injury) resulting from the above-mentioned entities and other disease states, e.g., sepsis [20]. The main elements involved in promoting kidney damage are dendritic cells, NK (natural killer) cells, macrophages, and proinflammatory cytokines. A critical role is played by the complement system, which can both protect and promote damage to the glomeruli [21].

Literature data suggest the contribution of innate immunity to TLRs in these processes. These receptors are a classic example of pattern recognition receptors (PRRs). Signals received by these receptors by recruiting specific molecules lead to activation of the transcription factors NF-κB (nuclear factor kappa-light-chain-enhancer of activated B cells) and IRF (interferon regulatory factor) and affect various elements of the host’s innate immune response [22]. TLR mechanisms are based on the ability to recognize twofold signals. The first one is based on the detection of pathogen-associated molecular patterns (PAMPs), while the second reads molecules related to damage to the body’s own cells (DAMPs; danger-associated molecular patterns) [23]. However, the origin of the signal, in the case of PAMP, is compounds of exogenous origin, while DAMP receives endogenous information. The result of receiving signals from both pathways is the effector reaction in which the production of costimulatory molecules and cytokines takes place. The location of TLRs is the cell surface or intracellular compartments, including ER (endoplasmic reticulum), lysosomes, and endosomes. It is pointed out that intracellular localization is important not only for ligand recognition but also for avoiding TLR contact with self-nucleic acids, which could cause autoimmunity. A high number of TLR receptors on cells of the immune system, such as monocytes, macrophages, dendritic cells, or lymphocytes, enables the thesis that they constitute a network allowing rapid cooperation of leukocytes and cells present at the site of infection or in the immediate vicinity of damaged host cells [23,24]. TLR synthesis begins in the rough endoplasmic reticulum, from where it goes to the Golgi apparatus and then to its destinations, in other words to the cell surface or to the intercellular compartments [22].

At present, there are 10 types of TLR receptors in humans and three additional types in mice, whereas others species may have more of these receptors (Table 1). Based on the amino acid sequence homology, TLRs occurring in vertebrates were divided into six subfamilies: TLR 1/2/6/10, TLR3, TLR4, TLR5, followed by TLR 7/8/9, and TLR 11 to the last 12/13 (Table 2). However, not all vertebrates have all types of receptors. PRRs have a specific structure in the form of transmembrane proteins, being an integral component of the cell membrane, in which the N-terminal part is responsible for ligand binding, whereas the C-terminal end is equipped with a signaling domain for IL-1 (TIR; toll IL-1 receptor), being part of the signal induction cascade for the production of anti-inflammatory mediators [23]. The transmembrane domain of TLRs contains about 20, mostly hydrophobic, amino acid residues. The N-terminal end (ECDs (extracellular domain) N-terminal ectodomains) is a glycoprotein of 500–800 amino acid residues. In their structure, we distinguish the presence of leucine-rich tandem repeats (LRRs), the number of which depends on the receptor type and ranges from 20–29 repeats (Table 1).

## 4. The Role of TLRs in Glomerulonephritis

Tissue damage glomerulonephritis predisposes many factors and individual characteristics. Most often, the first reveals hereditary predispositions in response to emerging environmental factors that can lead to a nephrogenic immune response. Then, there is a direct exposure to infectious etiological factors occurring in the environment (PAMP and DAMP), which may be subject to specific modification due to many epigenetic factors (such as physical exercise, microbes, environmental toxins, lifestyle, etc.) [62]. As a consequence of these changes, the innate immune system is activated as a result of interaction with TLRs present on circulating inflammatory cells (neutrophils, macrophages, basophils, and NK cells), as well as on resident glomerular cells and the complement system, which triggers a cascade of antigen-specific non-specific reactions [63]. These receptors are displayed on cells found inside the glomerulus (mesangial cells, monocytes, or dendritic cells) as well as in the renal interstitium (tubular epithelial cells, monocytes), where they interfere with potential ligands [63]. Part of the ligands, such as peptides, structural elements, or genetic material of both bacteria and viruses can be transmitted through the bloodstream to the inside of the nephron, in particular to the glomerulus. In the case of interstitial cells, in addition to the ligands of infectious origin, the potential ligands may also be fibrinogen, fibronectin, defensin 2, or necrotic cells (Figure 3).

In order to prevent over-activity of the immune system, TLRs are downregulated by numerous molecules and various mechanisms. Existing negative regulators target specific key molecules in TLR signaling, such as SOCS1 (suppressor of cytokine signaling 1), SOCS3 (suppressor of cytokine signaling 3), SARM (sterile α-and armadillo-motif containing protein), TANK (TRAF family member associated NF-κB activator), A20, and others [64,65,66,67,68,69]. Additionally, there are molecules that directly influence the inhibition of NF-kB and IRF-3 [70]. In addition, numerous mi-RNAs were discovered that affect the stability of mRNA encoding signaling molecules and mRNA for cytokines [69,71].

Activation of TLR releases the NF-κB transcription factor, which results in the production of inflammatory mediators (such as IL-1, IL-2, IL-6, IL-12, TNF-α (tumor necrosis factor alpha)), [72,73], and can cause glomerular damage. The next step is the conversion of the innate immune response that begins the antigen-specific reaction cycle. The transformation of the immune response includes several possible mechanisms, such as regulation of natural autoimmunity, conformational changes of epitopes, molecular mimicry, or the autoantigen complementarity phenomenon. TLRs are also required to activate the adaptive immune system by antigen-presenting cells that promote CD4 helper cell differentiation, B cell activation, and antibody production. Antibodies lead to the trapping of the circulating complex or the formation of in situ immune complexes that can activate both TLRs and complement components of the innate immune system [74]. CD4 Th1 and Th2 cells cause damage to the glomerular tissues indirectly, mainly through macrophages and basophils, whereas Th17 cells may directly mediate damage to kidney structures in particular diseases (Figure 4).

Various clinical situations affecting the kidneys, such as ischemic damage, toxic AKI, nephropathies secondary to diabetes mellitus, hypertension, or crystal deposition, are associated with aseptic inflammation caused, among others, by DAMP molecules [75,76]. These molecules can be released from dying parenchyma cells or during remodulation of the extracellular matrix. The presence of cells in the kidney capable of expressing TLR receptors makes it possible to initiate an immune response and inflammation [77,78]. In order to approximate the mode of action of these receptors in the disease state, their involvement in the most frequently diagnosed pathological conditions associated with glomerular dysfunction is presented in the further part of the material [8].

## 5. Biomarkers and Importance of the TLRs in Selected Glomerular Diseases

The literature classification divides the occurring nephropathies into two categories: Primary nephropathies, which are defined as those in which the systemic disease responsible for the condition cannot be established, and secondary nephropathies in which renal lesions appear as a result of other diseases accompanied by characteristic extrarenal symptoms. The first group includes diseases, such as: minimal change nephropathy (MCN), focal segmental glomerulosclerosis (FSGS), and membranous nephropathy (MN), which are also included in the nephrotic syndrome. The second group is primarily diabetic nephropathy and lupus nephropathy. In addition, the studies of our research group have resulted in the finding that the TLR-2 receptor may play an important role as a biomarker of primary non-proliferative nephropathies [79].

## 6. The Role of TLRs in Primary Non-Proliferative Nephropathies

### 6.1. Focal Segmental Glomerulosclerosis (FSGS)

FSGS is a diverse syndrome that arises after damage to podocytes for various reasons, some known and unknown. The sources of podocyte injury are diverse (circulating factors (primary FSGS), genetic abnormalities, viral infections, and medications) [80,81]. Most of the mutual interactions between these factors probably result in FSGS. There is a hypothesis about multistage pathogenic activation of autoimmunity in some forms of idiopathic FSGS [81]. Through the interaction of macrophages involved in kidney damage with many chemokines, the migration of monocytes to the site of damage occurs, which initiates the process of fibrosis. These macrophages also have the ability to self-spread and change into myofibroblasts that produce the extracellular matrix. Therefore, it can be assumed that excessive organ infiltration by monocytes and macrophages will cause an intensified fibrosis effect and, consequently, intensification of FSGS symptoms [82,83]. Currently, known biomarkers of FSGS are soluble urokinase-type plasminogen activator receptor (suPAR), soluble IL-2 receptor (sIL-2R), and ATP-binding cassette subfamily B member 1 glycoprotein-P (Figure 5). Damage to podocytes can release molecular patterns of proteins that are recognized by TLR as signals of danger. TLRs stimulate adapter proteins that activate a cascade of kinases, which amplify the signal and transmit it to the transcription factors regulating inflammatory genes. In the inflammatory microenvironment, the podocytes, acting as antigen-presenting cells, have a CD40 and CD80 receptor on their surface, thanks to which they capture antigens and present them to competent T-cells. However, in the case of abnormal expression of CD40 and CD80, they disorganize the cytoskeleton and filtration slit. In addition, CD40 can be identified as a foreign antigen, consequently leading to the production of anti-CD40 auto-antigen. Abnormal expression of CD40, CD80, and autoantibodies may lead to apoptosis of the podocytes, detachment of the podocytes from the glomerular basal membrane, proliferation of parietal epithelial cells, and attack on the glomeruli, and induction of segmental sclerosis [81]. In turn, other literature reports indicate the involvement of fibrinogen (Fg) in an inflammatory process mediated by the toll-like 4 receptor (TLR4) [84]. Fibrinogen is a protein that plays a proinflammatory role in vascular disorders, rheumatoid arthritis, glomerulonephritis, and certain cancers, e.g., myeloproliferative neoplasms [84,85]. Positive correlations have been noted between oxidative stress markers and, among others, fibrinogen, which may impact the course of several disorders [85]. Găman et al. [86] showed that obesity and diabetes are associated with increased levels of ROS (reactive oxygen species), accompanied by a simultaneous deficiency of antioxidants. The authors showed that the results of the free oxygen radical defense (FORT) and free oxygen radical defense (FORD) tests correlated with anthropometric/biochemical parameters in patients with obesity and diabetes. In studies carried out by Wang et al. [84], Fg has been shown to disrupt the actin cytoskeleton and induce apoptosis in podocytes via the TLR4-p38 MAPK-NF-κB p65 pathway in vitro and that co-expression of Fg and TLR4 is elevated in podocytes of Adriamycin-treated mice. It was also indicated that the level of fibrinogen in the urine may reflect the disease activity in patients with FSGS [84]. Literature data show that the use of synthetic small molecules lecinoxoids, which are inhibitors of TLR-2 and TLR-4, affects the activation and recruitment of monocytes in a rat model. The authors indicate [87] that the data demonstrate that targeting TLR-2-TLR-4 and/or monocyte migration directly affects the priming phase of fibrosis and may consequently perturb disease pathogenesis.

### 6.2. Minimal Change Disease (MCD)

MCD is one of the most common glomerular kidney diseases in children and a common cause of nephrotic syndrome in adults. This disease entity is characterized by an outbreak of edema, selective proteinuria, and a clinical response to glucocorticoid therapy, as T-cell mechanisms are involved in the pathogenesis of the disease [89]. The pathogenesis is due to abnormalities in the functioning of podocytes, with the latest literature data suggesting the hypothesis that there are two initiating events. First, there are changes in the cytoskeleton of podocytes and, second, there are regulatory changes in T-cells that exacerbate abnormalities in podocytes [89,90]. Currently known biomarkers in MCD are urine levels and podocyte expression of CD80 (B7.1), interleukin 13, serum levels and protease activity of circulating hemopexin, serum levels of soluble interleukin 2 receptor, and ABCB1 and glycoprotein-P (Figure 5). The development of MCD may be significantly influenced by the body’s innate immunity, in which TLRs are involved. Podocytes in the kidney glomeruli, due to their function and place of occurrence, are also equipped with the above receptors. In research conducted by Srivastava et al. [91], the presence of TLR receptors and their potential activity were checked on cell cultures stimulated with LPS (lipopolysaccharides) and the amino nucleoside puromycin (PAN). In the above studies, it was found that cultured human podocytes constitutively express TLR 1-6 and TLR-10 but not TLR 7–9. Quantitative analysis using the RT-PCR method indicated that LPS at various concentrations and to varying degrees increased the expression of TLR (1–6) genes, the adapter molecule MyD88, and the transcription factor NF-κB within one hour. LPS also caused elevated levels of IL-6, IL-8, and MCP-1 (monocyte chemoattractant protein-1) without exerting any effect on TNF-α, IFN-α, or TGF-β1 after 24 h. It has also been shown that an increase in TLR 1 expression may attenuate the effect of TLR-4 activation, which is thought to be an indirect factor in LPS-induced podocyte damage [91]. Moreover, the increase in TLR-1 expression by LPS suggests that LPS damage to podocytes is associated with an increase in TLR-1 levels and its specific endogenous ligands (heat shock protein, heparin sulphate, and fibronectin). These results allow the conclusion that the main TLR4 ligand, which is LPS, can induce the expression of the genes of many TLR receptors, and thus may lead to changes being induced in podocytes, which may be related to the loss of receptor selectivity and stimulation of receptor interaction in podocytes [91]. An additional possibility indicating the involvement of TLRs in the development of MCD is the increase in the amount of the CD80 receptor in podocytes, after stimulation with ligands for TLR-and TLR-4 receptors. TLR ligands are usually microbial products and can be combined with a well-known association of viral infections as a causative agent of minimal lesion disease [92,93].

### 6.3. Membranous Nephropathy (MN)

MN is a common cause of nephrotic syndrome in adults. Patients with MN usually develop severe proteinuria, edema, hypoalbuminemia, and hyperlipidemia [94]. It is the most common cause of idiopathic nephrotic syndrome in non-diabetic white adults. About 80% of cases are restricted to the kidneys (primary MN, PMN, idiopathic membranous nephropathy) and 20% are related to other systemic diseases or exposure (secondary MN) [95]. MN is associated with a pathological alteration of the glomerular basement membrane. This change is due to the build-up of immune complexes that appear as granular immunoglobulin (Ig)G deposits after immunofluorescence imaging and as electron-dense deposits of high electron density. Deposits of these immune complexes between podocytes and the basement membrane have a complex that attacks the complement membrane (C5b-9) [96]. The formation of glomerular sub-epithelial immune complex deposits in the IMN is mediated by specific intrinsic podocyte antigens and their corresponding autoantibodies in humans. These include compounds, such as neutral endopeptidase (NEP), type M receptor for secretory phospholipase A2 (PLA2R1), and type 1 7A thrombospondin (THSD7A) (containing domain 8–10) [94,95,96]. The above-mentioned markers constitute the core of the research into the pathogenesis of membranous nephropathy. However, there are reports of a genetic susceptibility to idiopathic membranous nephropathy. This type of study was conducted in a high-prevalence area in Taiwan [97]. In these studies, the association of the *IL-6, NPHS1 (nephrin), TLR-4, TLR-9, STAT4 (signal transducer and activator of transcription)* and *MYH (mutY DNA glycosylase),* genes with susceptibility to primary membranous nephropathy in Taiwan was established. In the case of the TLR4 receptor gene, the gene polymorphism indicated a significant single nucleotide difference in the rs10983755 A/G region (*p* < 0.001) and rs1927914 A/G (*p* < 0.05) between the control group and MN patients. In addition, the distributions of rs10759932 C/T and rs11536889 C/T polymorphisms differed significantly [97].

### 6.4. IgA Nephropathy (IgAN)

The development of IgA nephropathy consists of many mechanisms not yet fully understood. Literature data breaks down biomarkers for IgA nephropathy into a diagnostic and prognostic marker. The first group includes biomarkers detected in serum and urine, such as uromodulin, CD71, IL-6, complement components, and serum BAFF (B-cell activating factor) [98]. However, the group of prognostic markers includes urine kidney injury molecule-1, fractional excretion of IgG, soluble CD89, urinary angiotensinogen, and inflammatory cytokines (Figure 6). The above markers may indicate or predict the main cause of nephropathy development, which is the overproduction of anti-IgA complexes. One of the reasons for stimulating the body to produce IgA-related complexes may be signaling disorders associated with TLRs [99,100,101]. Ligands of bacterial or viral origin are recognized by toll-like receptors that trigger the process of chemokine release and recruitment of macrophages and neutrophils at the site of infection [102]. In numerous studies on IgAN pathogenesis, various types of receptors that could influence the development of this disease have been analyzed. These receptors were TLR3, mainly recognizing viral dsDNA [103]; TLR7 receptor [104]; binding to ssRNA viruses and TLR4 [105,106]; and binding of a variety of ligands, including LPS Gram-negative bacteria and DAMP (Table 2). Numerous observations of patients with diagnosed IgAN suggest the involvement of pathogens of viral origin, which is also confirmed by experimental models. There is an unexpected increase in the level of TLR4 activation, which is involved in the diagnosis of exogenous bacterial factors (LPS from Gram-negative bacteria, *Chlamydia pneumoniae*, HSP (heat shock proteins) proteins) and endogenous origin (HSP-60, additional fibronectin A domain, low-molecular LDL fractions, acid oligosaccharides hyaluronic acid, heparan sulphate), as well as factors derived from the breakdown of host cells [107,108,109,110,111,112,113]. To fully explain kidney damage in IgAN, it is necessary to fully understand the effects of TLR4 in the development of glomerulopathy, involving both glomerular cells and circulating leukocytes. Studies have shown that the administration of LPS activates TLR4 receptors on mesangial cells, and causes the release of chemokines (CXCs), which promotes neutrophil infusion and the development of glomerulonephritis [114,115,116]. In addition, the IFN-γ and IFN-α responses induced by TLR activation induce overexpression of the B-cell activation factor (BAFF) in dendritic cells, favoring the expansion of B-cells and increasing IgA synthesis [117,118,119,120,121]. It was also shown that in kidney biopsies of patients with IgAN, CD19+/CD5+ B cell infiltration is present, which in the progressive forms of this disease produce significant amounts of IFN-γ and IgA and are more resistant to apoptosis compared to cells obtained from healthy donors [122,123]. Moreover, Hitoshi Suzuki et al. [101] showed that there is an association between gene polymorphisms for TLR-9 and disease progression. Stimulation with ligands for TLR-9 led to the deterioration of kidney function in mice and influenced the shift of the balance towards Th1 lymphocytes. These findings led to the conclusion that activation of pathways related to this particular type of receptor may influence the severity of IgA nephropathy [101]. Moreover, Coppo et al. [124] showed that in patients diagnosed with IgA nephropathy, higher levels of TLR-4 in mononuclear cells and transcriptional mRNA were observed than in the control group. An important fact is that there is a statistical difference in the level of the above markers in patients with severe disease and those who do not have proteinuria and hematuria [124]. TLR-4 can be activated by many ligands, such as HLPs and LPS and DAMPs [106].

## 7. The Role of TLR Receptors in Secondary Nephropathies

### 7.1. Lupus Nephritis

Systemic lupus erythematosus (SLE) is one example of systemic autoimmune diseases. SLE relies on the loss of tolerance to autoantigens, which is caused by the malfunctioning of acquired immunity cells [109,125,126,127]. In the case of SLE, clinical studies indicate that the most common source of biomarker searches is a urine sample. Due to this, numerous proteins, such as cytokines, chemokines, complement proteins, adhesive molecules, and autoantibodies, have been identified as potential biomarkers of disease activity in cross-sectional studies (Figure 7) [128].

Literature data of recent years indicate a special contribution to the etiopathogenesis of this disease of innate immunity elements, mainly TLR. External and endogenous ligands may interact with TLRs present on monocytes, dendritic cells, and B lymphocytes, infiltrating the glomeruli, and resulting in increased cytokine secretion [129,130]. In addition, mesangial cells and other cells of the parenchyma express TLR1-4 and TLR6 receptors and secrete interleukins and chemokines [15,129,131,132,133,134]. Studies show that most deposits of immune complexes contain TLR agonists that have the ability to activate mesangial cells and contribute to the development of lupus nephropathy [135]. Pawar et al. [136] summarized the literature data, indicating that microbial nucleic acids can constitute a universal PAMP. As a result, it is possible to activate various mechanisms, such as lymphoproliferation, production of autoantibodies, type I interferon, secretion of numerous cytokines, and promotion of lupus development, in genetically predisposed individuals [136].

TLR2 and TLR4 are expressed not only in parenchymal cells but also in infiltrating neutrophils and mononuclear phagocytes, including macrophages and dendritic cells [137]. The HMGB1 (high mobility group box 1) protein, which binds DNA and the lupus autoantigen released under inflammation, can induce the activation of NF-κB in a TLR2-dependent and TLR4-RAGE-dependent manner in mononuclear phagocytes and neutrophils [138,139,140,141,142,143] as well as in mesangial cells [144]. Mesangial cells and podocytes in humans are characterized by the expression of TLR4 [145]. Mesangial cells isolated from mice with autoimmune diseases have significantly higher TLR4 expression and produce much more proinflammatory chemokines both after LPS stimulation and spontaneously [146]. Other studies also indicate that the necrotic cell debris-enhanced endogenous TLR ligands stimulate cytokine release by TLR2/MyD88 from mesangial cells, which implies the expression of TLR2 in different cell populations, and kidney-building structure [147,148,149,150]. Intracellular expression of TLR2 and TLR4 is multiplied in the kidneys of CD32b receptor-deficient mice suffering from glomerulonephritis associated with cryoglobulinemia [145,151,152].

### 7.2. Diabetic Nephropathy (DN)

One of the most serious complications for patients diagnosed with type I or II diabetes is diabetic nephropathy, which can be caused by both environmental and genetic factors [18,153]. The diabetic nephropathy is important to the inflammatory process in which, besides an increase in the activity of macrophages and overproduction of adhesion molecules of leukocyte cells, the proximal tubular kidney releases cytokine chemoattractant protein matrix to the interstitium, thereby contributing to the development of the disease [154,155,156]. Literature studies indicate that the greatest risk of diabetic nephropathy is the occurrence of hyperglycemia, which disrupts the proper functioning of the human body. On the molecular level, hyperglycemia is responsible for promoting the mitochondrial electron transport chain, which causes the formation of excessive amounts of reactive oxygen species (ROS) (through formation of the advanced glycation end products (AGEs) and activation of the polyol pathway, hexosamine pathway, protein kinase C (PKC), and angiotensin II). ROS occurring in the cell initiate or also intensify the formation of oxidative stress, which causes the intensification of inflammation and formation of fibrosis. Abnormalities in the lipid metabolism pathway, activation of the renin-angiotensin-aldosterone system (RAAS), as well as systemic and glomerular hypertension are also involved in the progression of this disease. Impairment of insulin signaling, an increase in growth factors and proinflammatory cytokines, and activation of the intracellular signaling pathway also play a role in the development of this disease [157]. Therefore, the currently known DN biomarkers focus on three areas: Detection of oxidative stress, the occurrence of inflammation, and activation of the RAAS system (Figure 8).

Literature data from recent years indicate that an important role in diabetic nephropathy etiopathogenesis may be played by signal transduction pathways that are dependent on TLR2, TLR9, and TLR4 receptors. However, it is not clear which receptor is more pathogenic [154,158,159,160]. Studies of many research teams indicate increased expression of TLR2 and the presence of endogenous ligands HMGB1 [161,162] and HSP70 [107,122,163], detected on the basis of research conducted on diabetic-induced rats. In addition, the expression of MyD88 and MCP-1, NF-κB, and infiltration of macrophages has been demonstrated [164,165,166,167]. High TLR2 expression was also observed in the glomeruli and renal tubules of people with diabetes. The results were confirmed in vitro in cultures of NRK-52E cell lines in which a high glucose concentration induced TLR2 mRNA expression [168,169,170]. The pathogenic role of TLR4 in diabetic nephropathy has also been found. In vitro studies have shown that activation of NF-κB and the expression of proinflammatory cytokines was reduced when TLR4 expression was silenced or its signaling was inhibited. During mouse studies, higher TLR4 activity and expression of the NF-κB p65 subunit in the kidney cortex of mice with experimental diabetes was demonstrated [171]. In vitro, researchers also observed increased TLR4 expression and proinflammatory cytokine synthesis when podocytes and adipocytes were exposed to both high glucose and NEFA (non-esterified “free” fatty acids), suggesting a key role of TLR4 in supporting inflammation in diabetic nephropathy [171,172,173]. In addition, another research team [174] showed an increase in the expression of TLR4 and signaling proteins for this receptor along with the activation of NF-κB, but not TLR2, in a mouse mesangium that was exposed to high glucose concentrations. It was also observed that hyperglycemia stimulated the expression of TLR4 in glomerular renal endothelial cells in a mouse model of type 1 and type 2 diabetes, which underlines the relationship of the discussed receptor with diabetic nephropathy [174,175]. In addition, clinical data indicate that in patients with type 1 diabetes, ligands, endotoxins, heat shock proteins 60 (Hsp60), and high mobility groups 1 (HMGB1) of both TLR2 and TLR4 [176,177] are increased. Increased expression of mRNA was also observed. TLR2, MyD88, and proinflammatory cytokines in leukocytes of patients with type 1 diabetes mellitus suggests that the inflammatory process is mediated by TLR2 [176]. However, in patients with type 2 diabetes and confirmed biopsy, diabetic kidney disease has been found to increase TLR4 expression on renal tubules as opposed to TLR2. In addition, in patients with type II diabetes and confirmed diabetic kidney disease, microalbuminuria, mRNA, and TLR4 protein were overexpressed 4 to 10 times more in glomeruli and tubules compared to the control group. However, both TLR2 and TLR4 expression was increased on monocytes in patients with type II diabetes [176]. Although the literature data indicate a key role of TLR2 and TLR4 in the pathophysiology of diabetic nephropathy, the participation of other receptors is not excluded. There is evidence to suggest the involvement of TLR3, TLR7, and TLR9 receptors in the pathogenesis of type 1 diabetes by destroying pancreatic islets induced by viral infection [114,178,179].

### 7.3. Acute Kidney Injury (AKI) to Chronic Kidney Disease (CKD) Development

The 2019 definition published by Ronco in The Lancet says that AKI: “is defined by a rapid increase in serum creatinine, decrease in urine output, or both”. The authors indicate that acute kidney injury accounts for up to 15% of the reasons for admission to hospital and occurs in as much as 50% of patients treated in intensive care units [180]. Therefore, it is argued that AKI is still associated with high morbidity and mortality [181]. There are many causes of AKI, including acute ischemia, analgesic nephropathy, sepsis, and severe glomerulopathies. Meta-analyses and cohort studies confirm the role of acute kidney injury in the development of chronic kidney disease and, as a complication of AKI, the more frequent need for chronic renal replacement therapy in the form of dialysis therapy [182,183,184]. The literature does not indicate clear causes and markers of the transition of acute kidney injury to chronic kidney disease, although it is known that this process involves numerous immune mechanisms. Inflammation, induced death, and fibrosis are likely contributors to the progression of AKI to CKD [185]. The renal epithelium plays an important role in promoting inflammation after damage by attracting leukocytes to the site of damage, which is seen as a strong role for TLR-4 [186]. The role of TLR2 and TLR4 in reperfusion after ischemic kidney injury is significant, where their expression increases significantly and enhances the proinflammatory response of tissues, with the participation of numerous cytokines and chemokines [187,188]. It is also noteworthy that the TLR-3 receptor system is involved in ischemia/reperfusion injury in the kidney. Studies have been carried out that showed rapid activation of these receptors, resulting in significant kidney damage, which was associated with elevated rates of apoptosis and necrosis in the renal tubes of mice [189]. Both TLR2 and TLR4 are also involved in sepsis-induced kidney damage [190], caused by Gram-positive and G-negative bacteria; nevertheless, the roles of single TLRs are differing [191]. There is also the TLR4-IL-22 pathway, which probably has regenerative functions in opposition to the above, and more research is needed on this topic [192]. After the action of a harmful factor, the repair process takes place, the element of which is sterile ignition [193]. In the first stage of inflammation, neutrophilic exudate appears, which over time is replaced by a monocytic-lymphocytic infiltrate [194,195]. The presence of monocytes is regarded as a mechanism promoting fibrosis and fibroblast proliferation [196]. The influx of cells of the immune system leads to cell apoptosis and the formation of a large amount of breakdown products and other substances that act as DAMPs, which can activate receptors, including TLRs [78,197]. It then promotes and strengthens the inflammatory response, attracting more cells of the immune system and subsequent fibrosis. Kidney infiltration by monocytes is recognized as a feature of chronic kidney disease [198] and the degree of monocyte infiltration correlates with the severity of kidney damage [199]. The above data indicate that it is the intensified process of fibrosis, mediated largely by the elements of the immune system, that can lead to permanent damage to the kidney function and the progression of AKI to CKD.

## 8. Conclusions

The innate immunity system in which TLRs participate, among others, cells, such as monocytes, macrophages, and NK (natural killer) cells, expressing TLR and lectin receptors are the body’s first line of defense. TLRs identify the pathogen based on the specific pattern of the molecule and therefore stimulate the immune system acquired, including T and B lymphocytes to fight pathogens [15]. Long-term continuous antigenic stimulation, increasing the work of TLRs, can lead to the development of serious diseases, organ specific or systemic. In an increasing number of studies, the role of TLRs has become more important. Recognition receptors are certainly of importance in the pathogenesis of diseases associated with nephropathy, for example, glomerulopathy, diabetic, lupus IgA, or FSGS [1,16]. However, further research is necessary to clarify the possible involvement of TLR in the development of other disease entities associated with kidney damage.

## Figures and Tables

**Figure 1 ijms-21-06712-f001:**
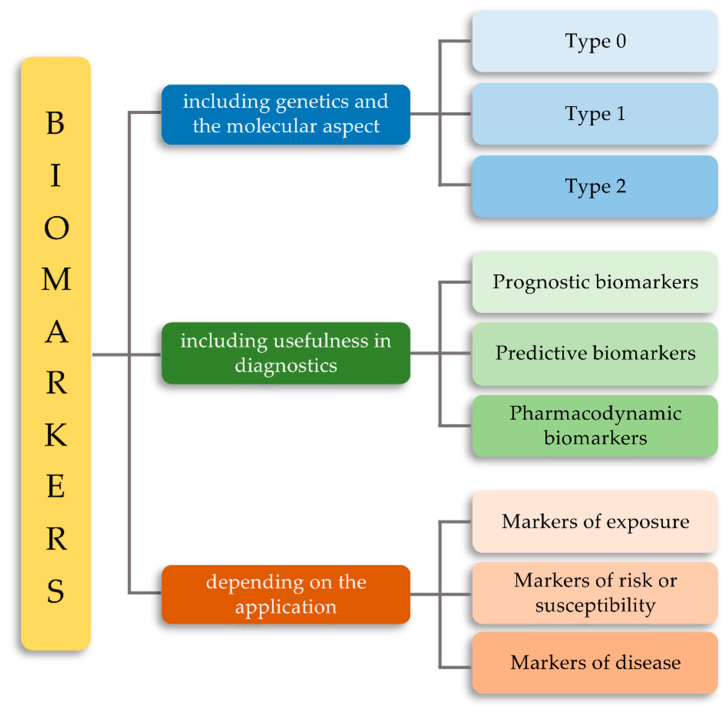
Classification of biomarkers due to the genetic and molecular aspect, the usefulness of biomarkers in diagnostic processes, and application.

**Figure 2 ijms-21-06712-f002:**
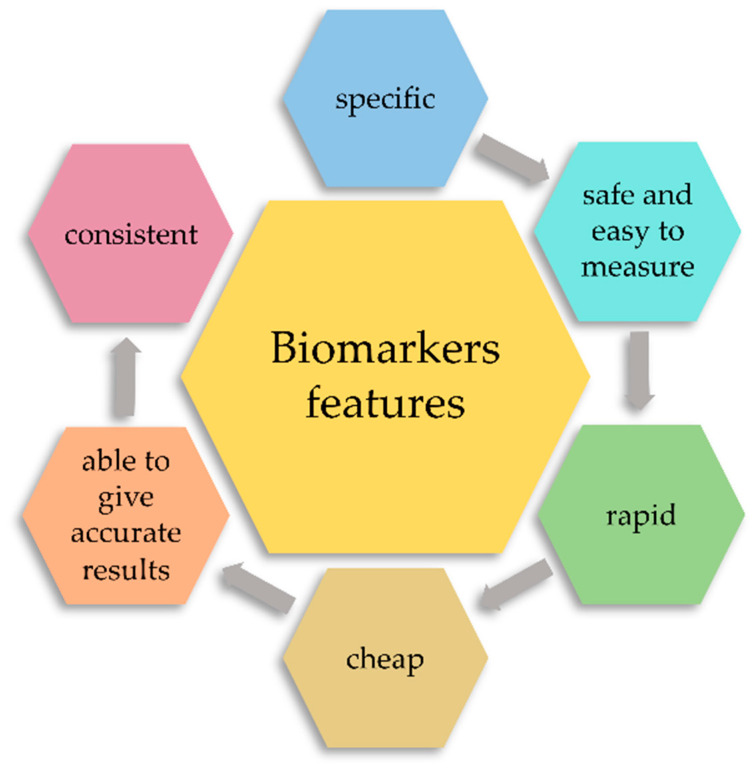
Classification of biomarkers due to the genetic and molecular aspect, the usefulness of biomarkers in diagnostic processes, and application.

**Figure 3 ijms-21-06712-f003:**
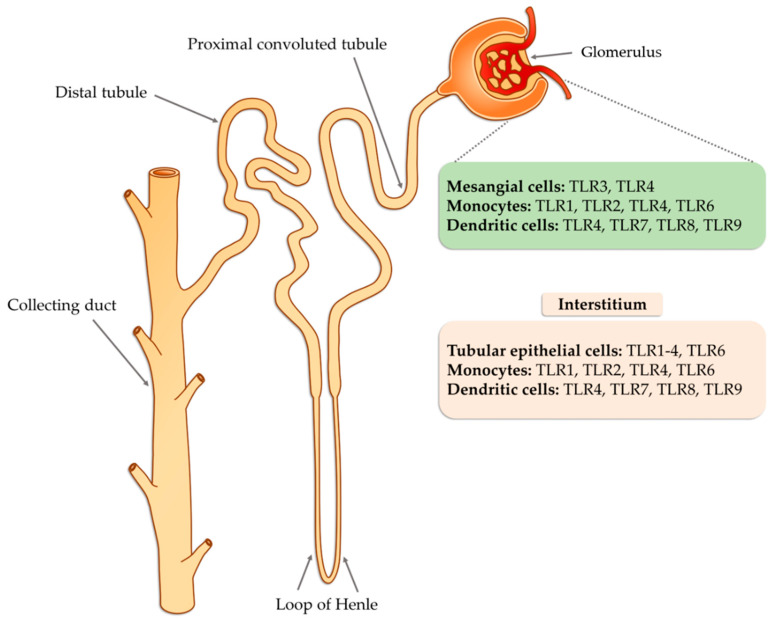
The potential occurrence of TLR receptors within the nephron (modified, based on [63]).

**Figure 4 ijms-21-06712-f004:**
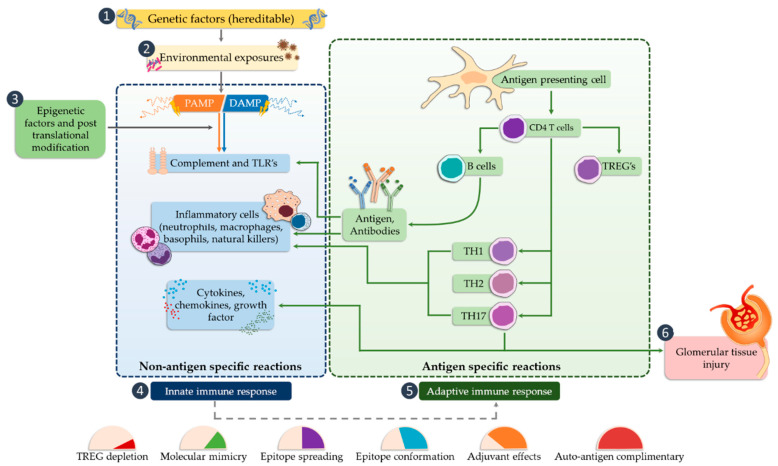
The potential occurrence of TLR receptors within the nephron (modified, based on [24]).

**Figure 5 ijms-21-06712-f005:**
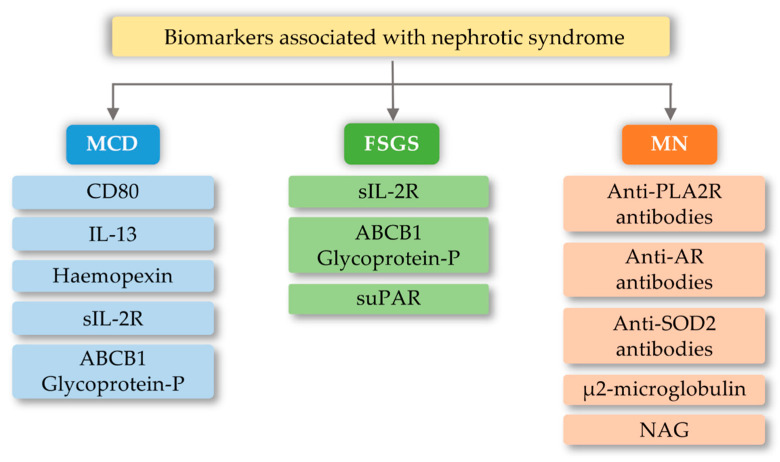
Biomarkers associated with nephrotic syndrome based on minimal change disease (MCD), focal segmental glomerulosclerosis (FSGS), and membranous nephropathy (MN) change based on [88]). CD80-Cluster of differentiation 80; IL-13-interleukin 13; sIL-2R-soluble IL-2 receptor; ABCB1 Glycoprotein-P-ATP-binding cassette subfamily B member 1 Glycoprotein-P; suPAR-soluble urokinase-type plasminogen activator receptor; PLA2R-M-type phospholipase A2 receptor; SOD2-manganese superoxide-dismutase 2; AR-Androgen Receptor; NAG-N-Acetyl-β-D Glucosaminidase.

**Figure 6 ijms-21-06712-f006:**
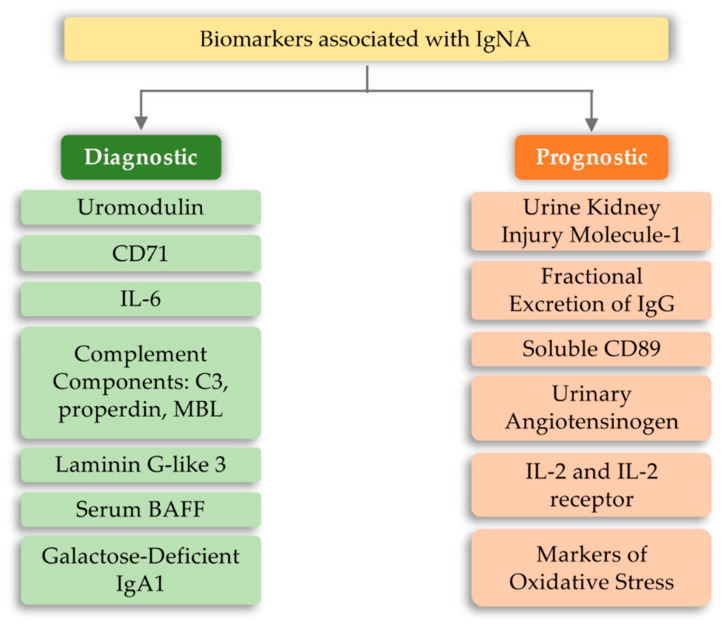
Biomarkers in IgA (immunoglobulin A) nephropathy.

**Figure 7 ijms-21-06712-f007:**
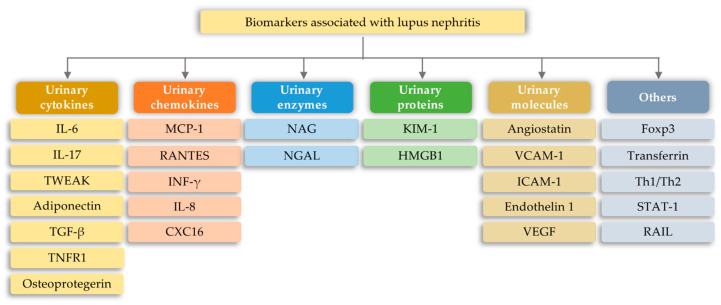
Biomarkers associated with lupus nephritis [128]; CXC16—C-X-C motif chemokine 16; FOXP3—forkhead box protein P3; HMGB1—High mobility group box 1; ICAM—Intercellular Adhesion Molecule 1; IL-6—Urinary Interleukin 6; IL-8—Interleukin 8; IL-17—Urinary Interleukin 17; KIM-1—Urinary kidney injury molecule-1; MCP-1—Monocyte chemoattractant protein-1; NAG—N-Acetyl-β-D Glucosaminidase; NGAL—neutrophil gelatinase-associated lipocalin; RAIL—Renal Activity Index for Lupus; RANTES—Regulated upon Activation, Normal T-cell Expressed, and Secreted; STAT-1—Signal transducer and activator of transcription 1; TGF-β—transforming growth factor beta; Th1—T helper 1; Th2—T helper 2; TNFR1—Tumor necrosis factor receptor 1; TWEAK—Urinary TNF-like weak inducer of apoptosis; VCAM—vascular cell adhesion molecule 1; VEGF—Vascular Endothelial Growth Factor.

**Figure 8 ijms-21-06712-f008:**
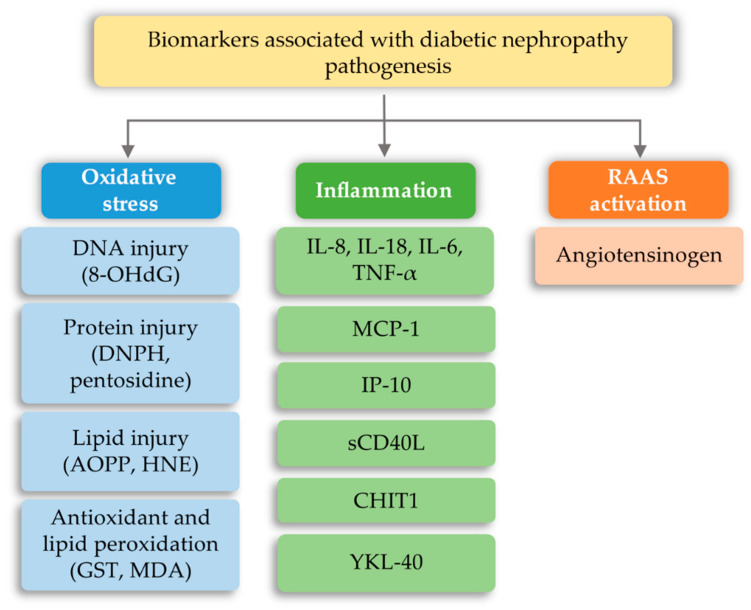
Biomarkers associated with diabetic nephropathy pathogenesis (based on [157]). 8-OHdG—8-hydroxy-2’-deoxyguanine; AOPP—advanced oxidation protein product; CHIT1—chitotriosidase; DNPH—2,4-dinitrophenylhydrazine; GS—Glutathione s-transferase; HNE—4-hydroxy-nonenal; IL-6—interleukin 6; IL-8—interleukin 8; IL-18—interleukin 18; IP-10—interferon-inducible protein-10; MCP-1—Monocyte chemoattractant protein-1; MDA—malondialdehyde; sCD40L—soluble CD40 ligand; TNF-α—Tumor necrosis factor alpha; YKL-40—cartilage glycoprotein 40.

**Table 1 ijms-21-06712-t001:** Characteristics of individual TLRs (toll-like receptors).

Name	Location of Coding Genes	Location in the Cell	The Number of Amino Acids	Molecular Weight (kDa)	Number of LLR	Reference
TLR1	Chromosome 4	Golgi apparatus, Phagosome, Cell membrane	786aa	90.31	19	[25,26,27]
TLR2	Chromosome 4	Phagosome	784aa	89.83	19	[25,28,29]
TLR3	Chromosome 4	Early endosome, ER	904aa	103.82	23	[25,30,31]
TLR4	Chromosome 9	Cell membrane, Early endosome	839aa	95.68	21	[25,32,33]
TLR5	Chromosome 1	No data	858aa	97.83	20	[25,34,35]
TLR6	Chromosome 4	Golgi apparatus, Cell membrane, Phagosome	796aa	91.88	19	[25,36,37]
TLR7	Chromosome X	Endosomes, Lysosomes, ER, Phagosome	1049aa	120.92	25	[25,38]
TLR8	Chromosome X	No data	1041aa	119.82	25	[25,39,40]
TLR9	Chromosome 3	Endosomes, Lysosomes, ER, Phagosome	1032aa	115.86	25	[25,41]
TLR10	Chromosome 4	No data	811aa	94.56	19	[25,42,43]
TLR11	Expression in mice	No data	926aa	105.83	10	[44]
TLR12	Expression in mice	No data	906aa	99.94	17	[45]
TLR13	Expression in mice	Endosomes	991aa	114.44	25	[46]

TLRs also differ in the cell site. TLR 1/2/4/5/6/10/11/12 receptors are located on the outer membrane of cells, while 3/7/8/9 receptors are located inside of them. Several literature reports have identified specific PAMP and DAMP ligands, which are bound by particular TLRs (Table 2).

**Table 2 ijms-21-06712-t002:** Location and ligands bound by TLR.

Name	Occurrence	Ligand PAMP	The Origin of PAMP	Ligand DAMP	Reference
**Extracellular**					
TLR1	Macrophages Neutrophils B lymphocytes Dendritic cells	Lipopeptides Soluble factors (lipoproteins)	Bacteria	No data	[47,48]
TLR2	Macrophages Neutrophils B lymphocytes Dendritic cells NK cells	Bacterial lipopeptides Teichoic acids, LAM Moduline, Glycolipids of bacteria, Porins, LPS,	Bacteria	Apolipoprotein CIII, Heparin sulphate, Hyaluronic acid, Hsp60, Hsp70, Peroxiredoxin	[47,48,49,50]
		Glycosinositolphospholipids	Protozoa, e.g., Trypanosoma cruzi		
		Zymosan	Fungi		
		Hemagglutinin	Measles virus		
		Protein	Herpesvirus		
		Hsp70 proteins	Host organism		
TLR4	Macrophages Neutrophils B lymphocytes Dendritic cells NK cells Treg cells	LPS	Bacteria	C-reactive protein, Fibronectin, Fibrinogen, Heparin sulphate, Neutrophil, Elastase, Angiotensin II, Hsp60	[48,49,50,51]
		Fusion proteins, Proteins present in the coating	Viruses, e.g., RSV virus		
		Taxol	Plants		
		Hsp60 protein Hsp70 protein A fragment of the A domain of fibronectin Hyaluronic acid oligosaccharide Fibrinogen Heparan sulphate	Host organism		
TLR5	Macrophages B lymphocytes Dendritic cells Treg cells	Flagellin	Bacteria (Gram-negative)	No data	[48,52]
TLR6	Macrophages Neutrophils Dendritic cells	Diacyl lipopeptides Lipoteichoic acids Zymosan	Bacteria Fungi	Versican	[47,48,49,50,53]
TLR10	Dendritic cells	No data	No data	No data	[54]
TLR11	Macrophages Dendritic cells	Flagellin Profilin	Bacteria Protozoa, e.g., Toxoplasma gondii	No data	[55,56]
TLR12	Dendritic cells	Profilin	Protozoa, e.g., Toxoplasma gondii	No data	[55]
**Intracellular**					
TLR3	Macrophages Neutrophils B lymphocytes Dendritic cells	Double-stranded RNA	Viruses	Own double-stranded RNA	[57,58]
TLR7	Macrophages Neutrophils Dendritic cells Treg cells	Single-stranded RNA Antiviral and anticancer compounds	Viruses Synthetic	Own single-stranded RNA	[48,59]
TLR8	Dendritic cells Treg cells	Single-stranded RNA Antiviral compounds	Viruses Synthetic	Own single-stranded RNA	[48,60]
TLR9	Macrophages Neutrophils Dendritic cells	Double-stranded DNA (containing unmethylated CpG sequences)	Bacteria, viruses and synthetic	HMGB1 Mitochondrial DNA	[48,49,50,61]

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
