# Peer review of "Toll-Like Receptor as a Potential Biomarker in Renal Diseases"

_ijms, 2020, doi:10.3390/ijms21186712_

Round 1
Reviewer 1 Report
Overall, this is a well-written manuscript. I have some minor suggestions:
- I don't think it is appropriate to equivalate biomarkers to "medical symptoms". A biomarker is "a component (analyte) of a human biological system (i.e., plasma, urine, etc.); or 2) a biomarker is a biological property (i.e., the mass concentration of X in plasma). Please see: https://www.degruyter.com/view/journals/cclm/51/9/article-p1689.xml
- The paper requires a minor polishing of the English language - I believe this can be corrected during copy-editing, but you must correct the figures as there are several misspelt words and there are missing spaces.
- The authors should further comment on the contribution of oxidative stress and low-grade chronic inflammation in the pathogenesis of glomerulopathy. Especially in obesity, diabetes and diabesity, the role of oxidative stress must be acknowledged. Furthermore, the authors should discuss the role of oxidative stress biomarkers and pro-inflammatory molecules - all of which can be evaluated by assays from capillary blood, serum, plasma or urine. Please see:
- https://www.ncbi.nlm.nih.gov/pmc/articles/PMC7243484/
- Moreover, fibrinogen is a marker of inflammation which is also elevated in certain cancers, e.g. myeloproliferative neoplasms as depicted it:
- https://doi.org/10.37358/RC.19.10.7581
- Also, a factor which can contribute to renal dysfunction in patients diagnosed with kidney disorders is the continuous polypharmacy these patients are subjected to. If considered worthy, the authors could also briefly discuss this topic to link their publication to everyday activity in the clinical ward. Please see:
- https://www.mdpi.com/1010-660X/55/8/436
I congratulate the authors for their interesting paper. Undoubtedly, a lot of effort has been invested in the writing of this manuscript.
Author Response
Dear Reviewer 1,
On behalf of the Authors of our review paper entitled “Toll like receptor as potential biomarker in renal diseases”, written by Mertowski S., Lipa P., Morawska I., Niedźwiedzka-Rystwej P., BÄ™bnowska D., Hrynkiewicz R., Grywalska E., RoliÅ„ski J., ZaÅ‚uska W., we would like to cordially thank for your review. We tried to follow all your suggestions and corrected the manuscript accordingly. Here are the point-by-point answers, all changes in the manuscript are marked in red.
1.Overall, this is a well-written manuscript. I have some minor suggestions: I don't think it is appropriate to equivalate biomarkers to "medical symptoms". A biomarker is "a component (analyte) of a human biological system (i.e., plasma, urine, etc.); or 2) a biomarker is a biological property (i.e., the mass concentration of X in plasma). Please see: https://www.degruyter.com/view/journals/cclm/51/9/article-p1689.xml
The definition of the biomarker has been rewritten, according to the proposed paper.
2. The paper requires a minor polishing of the English language - I believe this can be corrected during copy-editing, but you must correct the figures as there are several misspelt words and there are missing spaces.
All the typos we have found were corrected, also misspelt words in figures have been corrected.
3.The authors should further comment on the contribution of oxidative stress and low-grade chronic inflammation in the pathogenesis of glomerulopathy. Especially in obesity, diabetes and diabesity, the role of oxidative stress must be acknowledged. Furthermore, the authors should discuss the role of oxidative stress biomarkers and pro-inflammatory molecules - all of which can be evaluated by assays from capillary blood, serum, plasma or urine. Please see:
4.https://www.ncbi.nlm.nih.gov/pmc/articles/PMC7243484/
We tried to discuss the role of oxidative stress biomarkers, as suggested by the Reviewer.
5.Moreover, fibrinogen is a marker of inflammation which is also elevated in certain cancers, e.g. myeloproliferative neoplasms as depicted it:
6. https://doi.org/10.37358/RC.19.10.7581
- As suggested by the Reviewer, we have added the information described in the suggested paper.
7. Also, a factor which can contribute to renal dysfunction in patients diagnosed with kidney disorders is the continuous polypharmacy these patients are subjected to. If considered worthy, the authors could also briefly discuss this topic to link their publication to everyday activity in the clinical ward. Please see:
8. https://www.mdpi.com/1010-660X/55/8/436
According to the Reviewer suggestion, we have added information on the impact of continuous polypharmacy of the patients with renal disorders and T2D.
Again, we would like to thank you for your effort and time and we are hoping that our manuscript in its current form will fulfill the requirements of the IJMS.
Thank you for your time and consideration,
Sebastian Mertowski
Reviewer 2 Report
Mertowski and colleagues reviewed the current litterature on the role of TLRs in kidney diseases, and their possible implication as biomarkers. The review is well written and clearly accessible to non-experts (as myself). The load of information makes it quite instructive and it deserves to be published.
Based on the title I was hoping more information regarding the clinical link between TLRs and kidney diseases. For instance, what are the cellular and molecular consequences of polymorphisms in TLRs associated with kidney diseases? Is the expression or activity of (specific) TLRs altered in kidney diseases, in specific cell types…? How can this information improved kidney disease management or treatment in human (prognostic value? predictive value? markers?)?
From the manuscript it appears that most of our understaing of TLRs function in kidney diseases relies on mice models. How much of this information was confirmed in human? and successfully converted to patient care? In addition, is there some specific relationships between kidney diseases and specific TLRs? Is there an hypothesis to these specific relationships?
Again, the review is very nice but, to me, the title does not properly reflect its content. I believe authors could further improve the quality of the review by clarifying the points mentionned above.
Minor :
Lane 43-44: I would rephrase "Medical symptoms…"
Lane 272-273: Correct "citation".
Author Response
Dear Reviewer 2,
On behalf of the Authors of our review paper entitled “Toll like receptor as potential biomarker in renal diseases”, written by Mertowski S., Lipa P., Morawska I., Niedźwiedzka-Rystwej P., BÄ™bnowska D., Hrynkiewicz R., Grywalska E., RoliÅ„ski J., ZaÅ‚uska W., we would like to cordially thank for your review. We tried to follow all your suggestions and corrected the manuscript accordingly. Here are the point-by-point answers, all changes in the manuscript are marked in red.
1. Mertowski and colleagues reviewed the current literature on the role of TLRs in kidney diseases, and their possible implication as biomarkers. The review is well written and clearly accessible to non-experts (as myself). The load of information makes it quite instructive and it deserves to be published. Based on the title I was hoping for more information regarding the clinical link between TLRs and kidney diseases. For instance, what are the cellular and molecular consequences of polymorphisms in TLRs associated with kidney diseases? Is the expression or activity of (specific) TLRs altered in kidney diseases, in specific cell types…? How can this information improve kidney disease management or treatment in human (prognostic value? predictive value? markers?)? From the manuscript it appears that most of our understanding of TLRs function in kidney diseases relies on mice models. How much of this information was confirmed in humans? and successfully converted to patient care? In addition, are there some specific relationships between kidney diseases and specific TLRs? Is there an hypothesis to these specific relationships? Again, the review is very nice but, to me, the title does not properly reflect its content. I believe authors could further improve the quality of the review by clarifying the points mentioned above.
1.The issues raised by the second Reviewer are very prompt and we would be happy to answer those doubts, but according to the existing literature and studies the issues are still unsolved. Many of the data are based on mice models, and the information has not been yet converted to humans, that is why we decided to have in the name of the paper the word ‘potential’, to emphasize only the future possibilities of TLRs as biomarkers in renal diseases. We can only assume, based on the knowledge from the literature, that probably the strong link between a specific TLR and a renal disease will be questionable, as the TLRs’ ligands are dependent rather than specific on PAMP or DAMP. Possibly, a ‘cocktail’ of TLRs will be measured in future, to designate the status of kidney disease or its etiology. Depending on the factor contributing mostly to the disease, we can predict that different TLR may be involved, but possibly, as written in our paper and seen in different papers, TLR4 may be the strongest activated receptor among TLRs, due to its wide range of ligands that activate it. For this purpose, we submitted a project to the Polish National Science Center for the study of TLR4 genetic polymorphisms in patients with non-proliferative nephropathy, which is now being assessed by experts. Nevertheless, TLRs still require a lot of research, so at the moment we decided to refrain from adding far-reaching hypotheses to the text.
2. Line 43-44: I would rephrase "Medical symptoms…"
The sentence has been rephrased.
3.Line 272-273: Correct "citation".
We corrected the citation.
Again, we would like to thank you for your effort and time and we are hoping that our manuscript in its current form will fulfill the requirements of the IJMS.
Thank you for your time and consideration,
Sebastian Mertowski